# Application Prospects of Triphenylphosphine-Based Mitochondria-Targeted Cancer Therapy

**DOI:** 10.3390/cancers15030666

**Published:** 2023-01-21

**Authors:** Xiaoxia Cheng, Dong Feng, Junyu Lv, Xiaoman Cui, Yichen Wang, Qun Wang, Lei Zhang

**Affiliations:** 1School of Basic Medical Science, Henan University, Kaifeng 475004, China; 2School of Clinical Medicine, Henan University, Kaifeng 475004, China

**Keywords:** triphenylphosphine, delocalized lipophilic cations, mitochondria, cancer therapy, membrane potential

## Abstract

**Simple Summary:**

Although there are various treatments for tumors, chemotherapy is still one of the most dominant and irreplaceable treatment modalities nowadays. Unfortunately, chemotherapy is often accompanied by serious toxic side effects. Therefore, the search for new targets for antitumor drug action and the synthesis of new antitumor drugs has been a major topic of research in cancer therapy. With the development of medical science, excellent anti-tumor drugs need not only good therapeutic effects and less toxic side effects, but also preferably lower doses that can achieve therapeutic effects. Organelle-targeted antitumor agents can meet these requirements and have become one of the hot spots for antitumor drug research nowadays, such as mitochondria-targeted antitumor drugs. This review focuses on the application of triphenylphosphine (TPP) in mitochondria-targeted drugs and summarizes the common mitochondria-targeted carriers currently available, in the hope of providing a reference for the development of better mitochondria-targeted agents for tumor therapy.

**Abstract:**

Cancer is one of the leading causes of death and the most important impediments to the efforts to increase life expectancy worldwide. Currently, chemotherapy is the main treatment for cancer, but it is often accompanied by side effects that affect normal tissues and organs. The search for new alternatives to chemotherapy has been a hot research topic in the field of antineoplastic medicine. Drugs targeting diseased tissues or cells can significantly improve the efficacy of drugs. Therefore, organelle-targeted antitumor drugs are being explored, such as mitochondria-targeted antitumor drugs. Mitochondria is the central site of cellular energy production and plays an important role in cell survival and death. Moreover, a large number of studies have shown a close association between mitochondrial metabolism and tumorigenesis and progression, making mitochondria a promising new target for cancer therapy. Combining mitochondrial targeting agents with drug molecules is an effective way of mitochondrial targeting. In addition, hyperpolarized tumor cell membranes and mitochondrial membrane potentially allow selective accumulation of mitochondria-targeted drugs. This enhances the direct killing of tumor cells by drug molecules while minimizing the potential toxicity to normal cells. In this review, we discuss the common pro-mitochondrial agents, the advantages of triphenylphosphine (TPP) in mitochondrial-targeted cancer therapy and systematically summarize various TPP-based mitochondria-targeting anticancer drugs.

## 1. Introduction

Cancer is the leading cause of death worldwide and a significant barrier to increasing life expectancy [1]. The Global Cancer Statistics (2020) released by the World Health Organization’s International Agency for Research on Cancer (IRAC) show that 19.29 million new cancer cases and 9.96 million deaths were reported worldwide, with the number of people with cancer expected to reach 28.4 million by 2040, a 47% increase from 2020 [2]. Currently, chemotherapy is the main treatment for cancer, but most chemotherapeutic drugs are not specific. Therefore, they can also kill normal tissue cells. In addition, traditional drug molecules have disadvantages such as poor water solubility, low dissolution rate and low permeability leading to poor efficacy and adverse reactions [3]. Therefore, it is of great interest to explore ways to synthesize drugs with high anti-cancer activity and good targeting. To improve the targeting effect of chemotherapeutic drugs on tumors and increase the effective utilization of drugs, researchers have proposed the idea of “targeted drug delivery”, that is, to selectively deliver antitumor drugs to cancerous sites in the body by using carriers with special affinity or using the drugs themselves with strong specific affinity to a certain target site. This can reduce the toxic side effects on normal tissues and increase the concentration of drugs in cancerous tissues, resulting in high therapeutic effects [4].

Mitochondria, as semi-autonomous organelles, are the energy factories of the cell and are responsible for many important functions such as electron transfer, ATP synthesis [5] and metabolic pathways such as the citric acid cycle, gluconeogenesis, and fatty acid oxidation [6]. The mitochondrial membrane potential is the result of redox associated with Krebs cycle and is used as an intermediate form of energy storage by ATP synthase to make ATP. These transitions generate not only the electric potential (due to charge separation) but also the proton gradient, and together they form the transmembrane potential of the hydrogen ion [7]. The intracellular mitochondrial membrane potential and ATP levels fluctuate to a limited extent in response to normal cellular physiological activity, but always remain relatively stable. However, when these two factors are out of regulation, they may impair cell viability and lead to pathological consequences [8]. It follows that the unequal distribution of some key ions inside and outside the membrane, as well as the selective permeability to membrane movement, underlie the formation of membrane potential. In addition to mitochondria, organelles with membrane structures such as endoplasmic reticulum, Golgi apparatus and lysosomes in eukaryotic cells have biological membranes with different selective permeability to different ions and concentration differences between the two sides of the membrane. According to electrochemical principles, there should also be an electrical potential difference between the two sides of these organelle membranes and may be closely related to their physiological functions in the cell [9,10]. However, so far there is still a gap in the characterization of membrane potentials of organelles such as the endoplasmic reticulum, except for the mitochondrial inner membrane where a stable potential difference has been demonstrated on both sides. A very important reason may be that they are difficult to study directly by electrophysiological methods and to obtain results with higher resolution by the movement of potential-sensitive fluorescent dyes.

In addition, mitochondria also play an important role in various processes, such as apoptosis, oxygen sensitivity, cell differentiation and calcium metabolism [11]. Given that several diseases such as cancer, diabetes, neurodegenerative diseases, and ischemia-reperfusion injury are associated with mitochondrial dysfunction [12], there has been a growing interest in the mitochondria as drug targets in the last two decades. Many clinically approved drugs act directly on mitochondria to trigger cancer cell death. For example, a team of researchers screened within a library of FDA-approved small molecule drug compounds and identified 13 compounds that inhibit mitochondrial respiration and ATP synthase activity [13]. Among these compounds was nebivolol, a third-generation β1-adrenoceptor inhibitor. This study suggests that nebivolol can be used as a novel anticancer drug to treat cancer patients in addition to its original application in the treatment of hypertension and heart failure. Its anticancer mechanism is mainly through the inhibition of tumor angiogenesis and mitochondrial function, thus starving tumor cells and further inhibiting tumor growth. A number of studies have found that some drugs acting on intra-mitochondrial targets may have good efficacy. For example, protopine is an isoquinoline alkaloid from corydalis rhizoma with a wide range of biological activities including anti-inflammatory, antifungal, and antithrombotic. Now it has been shown that protopine can inhibit the growth of hepatocellular carcinoma through the mitochondrial apoptotic pathway, suggesting that mitochondria is a key target for tumor therapy [14]. As a result of the ongoing search for mitochondria-targeted drugs and the design of strategies to deliver drug molecules to mitochondria, a number of new terms have emerged, Mitocans being one of them. Mitocans is a term proposed by Ralph et al. in 2006 [15] to describe all drugs that can specifically act on mitochondria to inhibit cancer developement. Over the past 20 years, the development of Mitocans has yielded clear results. We summarize the classification of Mitocans according to the functional differences and the mitochondria properties they target in cancer cells rather than in normal cells. Thus, Mitocans include hexokinase inhibitors; thiol redox inhibitors; activators of mitochondrial membrane permeability transition pores; voltage-dependent anion channel/adenine nucleotide translocase (VDAC/ANT) targeting agents; inhibitors of Bcl-2 anti-apoptotic family proteins; electron transport chain/respiratory chain blockers; tricarboxylic acid (TCA) cycle targeting agents; mitochondrial inner membrane targeting agents; mtDNA targeting agents [15,16]. Nina et al. recently identified a new target, not directly targeting the mtDNA already present in the cell, but targeting the human DNA polymerase (POLRMT), to inhibit mitochondrial transcription which off the energy supply to tumor cells [17].

In this paper, we review the various pro-mitochondrial agents that have been developed, particularly the lipophilic cationic triphenylphosphine and discuss the advantages of TPP in mitochondrial targeting functions and the challenges associated with the application of TPP-based mitochondria-targeted anticancer drugs for oncology therapy and mitochondria-targeted drug delivery systems based on lipophilic cations.

## 2. Mitochondria-Targeted Agents

Mitochondrial targeting agents are compounds with specific physicochemical properties and reactivity. They can be small molecules that are lipophilic and positively charged, or peptide carriers or metal complexes such as ruthenium, iridium [18]. While they are diverse, they all have the same characteristic of having a high affinity for mitochondria. They can accumulate in the mitochondria without the aid of additional mitochondrial targeting systems [19]. Mitochondrial targeting belongs to organelle-level targeting, and the large transmembrane potential and mitochondrial protein entry mechanism are the two essential features used to target drugs. Therefore, drug targeting to the mitochondria is mainly achieved in two pathways: Lipophilic cation-mediated and mitochondrial protein import, in addition to mitochondrial targeting by linking nitroxide and cellular membrane penetrating peptides to drugs [20,21]. The first such compound, methyl-triphenylphosphonium (TPP) cation, was designed in 1969. This compound can selectively accumulate in mammalian cell mitochondria in response to larger membrane potentials [22]. Liberman et al. first used alkylated TPP cations as mitochondrial probes to study the relationship between mitochondrial membrane potential and oxidative phosphorylation and measure mitochondrial membrane potential [23]. Murphy and colleagues redesigned and refined the use of bioactive molecules bound to TPP in mitochondrial biology by selectively delivering drugs into the mitochondria through direct attachment of drug molecules to the TPP via alkyl chains or other covalent bonds [24,25]. Mitochondria contain about 1000–1500 proteins performing multiple functions. Mitochondrial proteins are derived from both the mitochondrial genome and the nuclear genome. More than 1000 proteins are encoded by nuclear genes and are synthesized as precursor proteins on cytoplasmic ribosomes [26]. These proteins are transported into the mitochondria mediated by targeting signals, hydrolyzed, folded, and are bound to cofactors to become mature proteins [27]. The mitochondrial targeting amino acid sequence signal (MTS) contained in this precursor protein can be combined with drugs or vectors to achieve mitochondrial targeting of drug molecules [28]. In addition, cell-penetrating peptides (CPP) can facilitate the entry of biomolecules such as proteins, peptides, SiRN and plasmid DNA as well as nanocarriers such as liposomes into the cell [29]. Shokolenko et al. combined CPP and MTS to facilitate the entry and mitochondrial targeting of the exonuclease III delivery system, further improving mitochondria-targeted transport [30].

Currently, the delivery of mitochondria-targeted drugs is mainly achieved by chemical ligation of active drugs by pro-mitochondrial agents or surface modification of nanocarriers. Among the commonly used mitochondria-targeting ligands include triphenylphosphine (TPP), rhodamine 19, rhodamine 123, diquathrin (DQA), F16, guanidinium salts, and mitochondria-penetrating peptides. Most of them are delocalized lipophilic cations (DLCs), and their lipid solubility enables them to cross cellular and mitochondrial membranes, while the positive charge they carry allows them to enter the mitochondrial matrix in the presence of the mitochondrial membrane potential, with mitochondrial targeting effects. The structure of some DLCs is shown in Figure 1. The discovery of such compounds dates back to the late 1970s, when scientists stained surface antigens with rhodamine attached to antibodies and surprisingly found that serpentine subcellular organelles in cells were also stained (later found to be mitochondria) [31]. In a subsequent series of studies, rhodamine 3B, a rhodamine derivative, was found to be the main component for staining organelles, and rhodamine reached cellular mitochondrial sites after 15 min of cell staining with a concentration of 0.5 µg/mL rhodamine 3B. Two rhodamine 123 and 6G designed as laser fuel by Kodak et al. were also found to have very good staining effect on living mitochondria [32]. In addition, rhodamine 19, which is similar to rhodamine 123 was considered as a potential mitochondrial targeting moiety and was used to replace TPP on TPP-drug adducts, and the new adducts exhibited good mitochondrial targeting properties, demonstrating that rhodamine 19 has good mitochondrial targeting ability [33]. However, other rhodamine derivatives such as Rhodamine B, 110, and 116 have not been found to have good mitochondrial targeting effects [32]. The study of rhodamine revealed that all rhodamine-like derivatives that can exclusively aggregate onto cellular mitochondria belong to the category of DLC [34]. They have esterified carboxyl groups and a positive charge that spans the entire molecule, so they do not affect the inherent lipophilicity of the dye, thus uncovering the secret that DLCs can target cellular mitochondria. This suggests that the DLCs targeting to mitochondria is strongly related to its structure. DLCs have two structural features in common [35]: (1) They consist of a hydrophilic charged group connected to a hydrophobic group by a chemical bond; (2) their π-electron cloud extends to a density of more than three atoms rather than being confined to the region between the heteroatom and the neighboring carbon atom, and this electron delocalization phenomenon makes the whole molecule positively charged.

Dequalinium chloride (DQA), which consists of two cationic quinoline groups linked by a 10-carbon alkyl chain, is an off-domain lipophilic cationic molecule with a mitochondrial targeting effect. This compound can inhibit the proliferation of a variety of cancer cell lines in vitro and also exerts antitumor activity in vivo [36,37]. DQAsomes are vesicle-like aggregates with sizes in the range of 70–700 nm in diameter formed by DQA and are highly stable. More importantly, they have the property of aggregating into mitochondria after penetrating the lipid bilayer. Therefore, DQAsomes could be an effective class of mitochondria-targeted nanocarriers for the development of mitochondria-targeted antitumor agents [38]. For example, Cheng et al. investigated the anti-human colon cancer effect of DQAsomes-based mitochondria-targeted delivery of paclitaxel in nude mice and showed that it significantly inhibited the growth of human colon cancer cells COLO-205 in nude mice [39]. Other studies also used DQA in combination with the doxorubicin to achieve mitochondria-targeted delivery of doxorubicin, which mainly accumulated in MCF-7/ADR cells causing high cytotoxicity [40]. F16 acts as an off-domain lipophilic cationic ligand with mitochondrial targeting effects. It was originally named F16 by Fantin et al. who found that it significantly inhibited the proliferation of mammary epithelial cells as well as a variety of mouse mammary tumor and human mammary tumor cell lines [41]. F16 is inherently cytotoxic, targeting tumor cell mitochondria to induce apoptosis and inducing tumor cell necrosis in the presence of anti-apoptotic Bcl-2 protein overexpression, with dual efficacy in killing tumor cells [42]. Additionally, examples of its use as a mitochondrial targeting ligand include the study by He Huan et al., where F16 was chemically linked to a borodipyrrole methylene fluorescent dye to prepare FPB, a bifunctional mitochondrial targeting anticancer agent with optical detection and selective anticancer activity [43]. There are also guanidinium salts, mitochondrial penetrating peptides, and mitochondrial targeting sequences that can be used as pro-mitochondrial agents to deliver drug molecules into the mitochondria due to their specific physicochemical properties or specific biological reactivity, which will not be discussed here [36,37,38].

## 3. Advantages of TPP with Targeted Tumor Cell Delivery

The first report of a mitochondria-targeted coupling using TPP was in 1995, when Liberman et al. used alkylated TPP cations as mitochondrial probes to explore mitochondrial oxidative stress processes [23]. In 1999, Smith’s group was the first to synthesize Mito-vite E, an antioxidant targeting mitochondria [44], and demonstrated that targeting antioxidants to mitochondria could effectively prevent mitochondrial oxidative damage. Subsequently, MitoQ10, a coenzyme Q10 and TPP conjugate [45], and MitoSOD, a SOD cofactor mimetic and TPP conjugate [46] were developed and found to provide selective protection against oxidative stress damage in tissues. In 2002, Fantin et al. synthesized F16-TPP, a compound with mitochondrial targeting and fluorescence properties, by combining TPP with F16 molecules via amidinium ions, which aggregated in large quantities in the mitochondria thereby killing a variety of tumor cells [41]. Structural modification of existing antitumor drugs is also an important approach for designing targeted mitochondrial antitumor drugs using TPPs. For example, PDT-PAO is an organic arsenic with anticancer efficacy against leukemia, lymphoma, and solid tumors [47]. Fan et al. designed and synthesized the mitochondria-targeting precursor PDT-PAO-TPP [48]. When conjugated with TPP, it showed enhanced tumor inhibition ability. In detail, PDT-PAO-TPP can inhibit the activity of pyruvate dehydrogenase complex, leading to the inhibition of ATP synthesis and impaired thermogenesis. In addition, inhibition of respiratory chain complexes I and IV by PDT-PAO-TPP promotes cellular respiratory dysfunction and ROS production. These results led to the toxicity of PDT-PAO-TPP to tumor cells. Adriamycin (DOX), as a broad-spectrum antitumor drug, exerts its effect mainly by entering the nucleus of cells and inhibiting the synthesis of DNA [49]. However, with its long-term use, it may stimulate cancer cells to overexpress p-glycoprotein(P-gp), activate nuclear DNA repair, increase glutathione transferase activity, which eventually leads to the development of drug resistance [50]. When drug resistance is developed, P-gp can use the energy of ATP hydrolysis to actively excreting the drug so that most DOX cannot reach the nucleus to act, and cytotoxicity is thus reduced [51]. If DOX can be delivered to mitochondria, interfere with mitochondrial DNA replication and affect cellular redox responses, it will be possible to kill drug-resistant tumor cells and thus reverse drug resistance. Since TPP has a high affinity for mitochondria, DOX can be linked to TPP to obtain TPP-DOX complexes with mitochondrial targeting. It was shown that the chemical linking of DOX to TPP showed good mitochondrial targeting, induced apoptosis and overcame tumor cell resistance through the mitochondrial pathway [52]. In conclusion, small-molecule compounds modified by TPP can achieve mitochondrial targeting. They can be applied in many fields and they therefore represent a reliable approach for modifying antioxidants and antitumor drugs. Besides, researchers have designed and synthesized many specific fluorescent probes using TPP, which can be used for drug action targets or detection of relevant indicators in mitochondria [53]. Currently, the combination of nanocarriers with TPP can target encapsulated small molecule drugs to mitochondria, which is gradually becoming a key concept in research.

### 3.1. TPP-Based Mitochondrial Targeting System Preferentially Delivers Drugs into Tumor Cell Mitochondria

Triphenylphosphine, as the most widely used mitochondria-targeting lipophilic cationic ligand, consists of a positively charged phosphorus ion linked to three benzene rings, the latter of which makes it highly lipid soluble. The phenyl group is spatially blocked to protect the phosphorus atom from solubilization. In addition, the positive charge on the phosphorus atom in the structure can be delocalized to the three benzene rings to form a delocalized positive charge, which promotes TPP to cross the lipid bilayer and target the inside of mitochondria [54]. TPP has been used for the measurement of mitochondrial membrane potential and its behavior and mitochondrial interactions are well defined [23]. Mitochondria provide the cell with the required ATP, and the proposed mechanism of ATP production is the chemiosmotic hypothesis developed by Mitchell et al. in 1961. When high-energy electrons are passed along the respiratory chain, the energy released causes protons to be pumped from the matrix side of the mitochondrial inner membrane to the intermembrane space [55]. In this process, not only the membrane potential (ΔΨm: due to charge separation) but also the proton gradient (mitochondrial pH gradient, ΔpHm) is generated. Together they form the transmembrane potential of H+ [7]. This better clarified why compounds targeted to mitochondria that are weak acids or bases may undergo differential protonation in the cytosolic and mitochondrial compartments due to differences in their pH. TPP-based compounds can be rapidly taken up by mitochondria driven by high mitochondrial membrane potential. In order to reach the mitochondria of intact cells, compounds must cross both the plasma and mitochondrial membranes. Fortunately, in both cases, the membrane potential is negative, which allows TPP compounds to accumulate progressively in the cytoplasm of the cell and then in the mitochondria. At equilibrium, the ion concentrations on both sides of the charged membrane can be described by the Nernst equation, while for a single-charged cationic species accumulating in the space enclosed by the membrane with a negative internal potential and a temperature of 37 degrees, the Nernst equation can be simplified as: ΔΨ (mV) = 61.5 xlog_10_^Cin/Cout^ [56].

As shown in Figure 2, for every 61.5 mV of membrane potential at 37 degrees, the uptake of TPP cations by mitochondria increases approximately 10-fold, and the potential of the cytoplasmic membrane is typically 30–60 mV compared to that of the extracellular compartment with a 5–10-fold increase in intracellular TPP ion concentration. The mitochondrial membrane potential is in the range of 150–180 mV, with a further 100–1000-fold increase in TPP in the mitochondrial matrix. The membrane potential of all cells and mitochondria may contribute to the accumulation of TPP cations in mitochondria, which also raises concerns about the cytotoxicity of normal cells. Fortunately, the mitochondrial membrane potential of tumor cells is usually 60 mV higher than that of normal cells because cancer cells have a stronger differentiation capacity than normal cells and require more energy from mitochondria for cell growth [57]. Chen et al. [58] examined more than 200 cell lines, including adenocarcinoma, metastatic carcinoma, squamous epithelial carcinoma and normal epithelial cells, and found that the mitochondrial membrane potential of tumor cells was higher than that of normal epithelial cells, and only 2% of cells did not follow this pattern [57]. This implies that TPP based on the drive of mitochondrial membrane potential can preferentially target antitumor drugs to the mitochondria of tumor cells and induce apoptosis. Thus TPP-based targeted drugs can selectively eliminate cancer cells.

TPP as a mitochondrial targeting ligand has certain advantages over other small molecule mitochondrial delivery strategies. TPP is relatively stable in biological systems, is both hydrophilic and lipophilic, relatively simple to synthesize and purify, has low chemical reactivity to cellular components and no light absorption or fluorescence emission in the visible and infrared spectral regions [56]. In addition, TPP is more likely to carry drug molecules across the membrane barrier and is safer than other lipophilic cations.

### 3.2. TPP Can Carry Drug Molecules across the Membrane Barrier

The lipid bilayer membrane structure creates a natural barrier around the cell and mitochondria, preventing the free passage of water-soluble substances on both sides of the membrane. The passage of drugs with lipophilic cations as carriers through the inner mitochondrial membrane is a multistep process [59], as shown in Figure 3. First, the lipophilic cation binds to the interstitial side of the mitochondrial membrane and is embedded in the phospholipid layer, and then is transferred to the matrix side of the membrane. Finally, it separates from the matrix side of the membrane and enters the mitochondrial matrix. The energy barrier to transfer of lipophilic cations through phospholipid bilayers is usually associated with the transfer of membrane-bound compounds from one side of the membrane to the other (second step in Figure 3) [60]. The lipophilic cation passes easily through the lipid bilayer due to its lipid solubility. Within a certain range, the higher the lipophilic cation’s lipid solubility, the easier it is to pass through the phospholipid bilayer, resulting in a more efficient transport of the carried drug and higher selectivity for the drug. Wang et al. [61] summarized the cLogP (calculated LogP: calculated by Chemdraw2014.) values of some common lipophilic cations. LogP is the Log value of the ratio of the concentration of a compound in the organic phase to its concentration in the aqueous phase in the neutral state. Higher LogP values imply that the compound is more lipophilic. The cLogP of TPP cation is 6.29. It is much larger than other cations such as indole cation (4.31) and rhodamine (4.93). The high cLogP of TPP explains why it passes more easily through the lipid bilayer. Another reason for allowing TPP to cross biological membranes more easily is because of the low activation energy required for its transfer from aqueous to hydrophobic environments, which may explain why other hydrophilic cations such as sodium or potassium cations cannot cross biological membranes without the facilitation of ion channels or carrier proteins. It has been reported that the activation energy required for lipophilic cations to enter biological membranes is related to Q (charge per mole of cation), R (ionic radius) [62]. Given that most lipophilic cations contain one positive charge per molecule, the activation energy required for cation entry into the membrane is inversely proportional to the ionic radius (R). Due to the three hydrophobic phenyl groups, the TPP cation has a relatively large ionic radius and is strongly hydrophobic, making the activation energy required for TPP to penetrate biological membranes significantly lower than that of other lipophilic cations.

### 3.3. Safety of TPP Compounds

High accumulation of lipophilic cations cause toxicity in the mitochondria. While they are taken up by mitochondria through the same mechanism, their mitochondrial toxicity mechanisms are different. For example, ammonium chloride and benzothiazole inhibit the NADH-ubiquinone reductase of respiratory complex I. Rhodamine 123 affects normal mitochondrial function by inhibiting ATP synthase [63,64]. Furthermore, MKT-077 is among the first delocalized lipophilic cations evaluated in clinical trials that showed anticancer activity but showed nephrotoxicity in phase I clinical trials, implying that the drug is not feasible as a mitochondria-targeted agent in cancer therapy. This may be related to its non-specific induction of mitochondrial membrane perturbation and non-specific inhibition of mitochondrial respiratory enzyme activity on the membrane by MKT-007 [65]. F16 is a newly discovered fluorescent delocalized lipophilic cation with anti-proliferative ability, which exhibits cytotoxicity associated with the induction of mitochondrial permeability transition pore opening, leading to mitochondrial dysfunction [41]. While F16 can selectively accumulate in the mitochondria and exhibit potent toxicity in tumor cells, this is not sufficient to make this compound a targeting agent for drug development, as structural modifications of F16 compounds cannot effectively inhibit cancer development [66]. However, it was found that TPP-based compounds (e.g., TPMP, MitoQ, Mito-Vit E) can be administered at high doses for long term without causing significant damage to organs such as the heart, liver and kidneys, and this was confirmed in rodent models [67]. After researchers fed these compounds to mice for several weeks, they found that the concentration of cations in the mitochondria tended to be constant. At this point, the concentration of cations in the mitochondria would be hundreds of times higher than the concentration in the blood, and the rate of oral absorption of these compounds would match the rate of excretion into the urine and bile. This means that the accumulation of TPP-based compounds in mitochondria is in dynamic equilibrium. In addition, oral administration of high doses of TPMP and Mito-VitE was well tolerated in mice (30 and 60 mg/kg/day, respectively). Currently, MitoQ and SKQ1 are in clinical trials and neither compound has shown any systemic toxicity when used at pharmacologically relevant doses [68,69]. MitoQ is currently commercialized and sold in stores as a skin care product. While TPP conjugated to stearoyl residues (STPP) has been shown to contribute to mitochondrial membrane permeability and induce instability and alterations in lipid bilayer properties [70], TPP cations exhibit cytotoxicity only at extremely high concentrations. This implies that the emergence of cytotoxicity in TPP-based compounds may be due to the drugs carried, rather than the TPP itself.

## 4. TPP-Based Mitocans

TPP-based mitochondria-targeted anticancer drugs mainly target tumor cells with high membrane potential and deliver the drug to the tumor cell mitochondria to kill or treat them. We classify the current use of TPP as a mitochondrial targeting agent into two categories (as shown in Figure 4): (1) TPP directly conjugated with drug molecules and (2) TPP-modified targeted mitochondrial nanosystems.

### 4.1. Direct Affixation of TPP to Drug Molecules

Based on the literature, we classified the anticancer drugs directly conjugated with TPP for mitochondrial targeting into four categories, as shown in the Figure 5: (1) Coventional cytotoxic agents; (2) natural drug molecules; (3) drugs initially used outside of cancer treatment; and (4) TPP derivatives.

#### 4.1.1. Traditional Cytotoxic Drugs

Several commonly used cytotoxic drugs such as cisplatin, doxorubicin (DOX) and tamoxifen have been conjugated to TPP. In addition, some of these derivatives have been further linked to nanocarrier systems and tested for their efficacy [62,63,64]. Cisplatin is a widely used anti-cancer drug for the treatment of many malignant tumors. The structure of cisplatin is simple and the mechanism of anti-cancer is well understood. Cisplatin is highly electrophilic compound so it can potentially react with nucleophilic sites of both nuclear DNA both DNA-RNA-protein of mitochondria) [71]. However, the clinical efficacy of cisplatin is severely limited by drug resistance, mainly due to reduced drug accumulation and increased cellular self-healing capacity [72,73]. In addition, cisplatin is not selective for tumor cells and has serious side effects. Studies have shown that cisplatin bound to TPP preferentially accumulates in mitochondria, bypassing nuclear DNA and entering the mitochondrial genome to affect cell functions. This approach overcomes drug resistance to cisplatin [74]. Researchers measured the distribution of cisplatin in the major organs of mice and found that most of the platinum accumulated in the liver and kidneys. However, TPP-cisplatin accumulated more platinum in lung and tumor tissues compared to cisplatin, which is less toxic to normal cells and has good compatibility. In addition, TPP-cisplatin has been implicated in DNA damage and also in other anticancer pathways, such as glycolysis inhibition, mitochondrial bioenergetics and cytochrome C release [75]. DOX acts as an antibiotic antineoplastic drug that inhibits the synthesis of RNA and DNA. It can treat various tumors but is causes strong side effects. In addition, tumor cells are likely to develop resistance to DOX [76]. It has been shown that conjugating TPP with DOX-carrying nanoparticles can cause severe acute cytotoxicity in prostate tumor cells [77,78]. This is because when DOX is delivered to the mitochondria, it can produce toxicity by interacting with mitochondrial DNA and can also trigger the generation of reactive oxygen species (ROS) by tumors above the threshold at which it activates proteins involved in apoptosis [51]. TPP-DOX has been reported to increase ROS production and toxicity in mammary tumor cells when combined with hyaluronic acid. In addition, both TPP-DOX and hyaluronic acid (HA)-TPP-DOX inhibited tumor cell growth in a zebrafish model with tumors, and both had no significant side effects [79]. However, HA-TPP-DOX showed better anti-tumor effects than TPP-DOX and could control the release of TPP-DOX and improve the sensitivity of anti-tumor cells [52,80]. Tamoxifen has been in clinical use for more than 40 years. It is the first drug approved by the FDA for breast cancer prevention in high-risk women, reducing the risk of breast cancer by up to 40% [81]. TPP-tamoxifen is a novel mitochondria-targeting derivative of tamoxifen. Several researchers have found that, unlike tamoxifen, TPP-tamoxifen effectively inhibits experimental Her2 overexpressing tumors without systemic toxicity. This is mainly due to the fact that TPP-tamoxifen can selectively accumulate in the mitochondria of breast cancer cells and overexpressing HER2 tumors, leading to the disruption of the supercomplex, inhibition of complex I, increased production of reactive oxygen species and dissipation of membrane potential, thus showing toxicity to tumor cells [82].

#### 4.1.2. Natural Drug Molecules

Many early cancer chemopreventive agents are limited to nutrients, such as vitamin C and vitamin E [83]. However, in recent decades, non-nutritional chemical phytoconstituents have also received increasing attention. Natural products are an important source for the discovery of pharmacologically active small molecules, and the antitumor activity of natural products coupled with TPP is significantly increased. Honokiol (HNK) is a key bioactive compound in Magnolia officinalis bark extract that inhibits mitochondrial respiration, decreases ATP levels, and increases ROS production, and exhibits anti-tumor and anti-metastatic activities in both in vivo and in vitro assays [84]. Mitochondria-targeted HNK (Mito-HNK) promotes the accumulation of mitochondrial HNK, and its ability to inhibit cell proliferation, suppress mitochondrial complex I, inhibit mitochondrial STAT3 phosphorylation, and stimulate reactive oxygen species production is 100-fold higher than that of conventional HNK [85]. In addition, in vivo experiments showed that Mito-HNK showed no significant toxicity in mice. Similar to Honokiol, glycyrrhetinic acid is a pentacyclic triterpenoid extracted from plants. While it has antitumor activity, its use is limited by the low bioavailability and poor water solubility. By combining with TPP, this compound can specifically target mitochondria to induce cell cycle arrest and thus inhibit cancer cell proliferation and migration [86]. Andrea et al. synthesized two Psoralen derivatives, PAPTP and PCARBTP, which showed good targeting ability of the drug induced by TPP [87]. Both drugs have been shown to be promising chemotherapeutic agents that selectively eliminate cancer cells in vitro and in vivo in tumor models, including melanoma and pancreatic ductal adenocarcinoma (PDAC). Most importantly, they have no significant effect on healthy tissues and cells, including immune organs in mice and humans [88]. Polyphenols are a large group of natural compounds with a variety of important biomedical activities, at least in vitro. Their potential relevance to healthcare in terms of cardiovascular [89] and neurological protection [90], cancer prevention and treatment, and reduction of chronic inflammation has been widely demonstrated [91]. Quercetin and resveratrol are both widely used model polyphenols. Quercetin belongs to the flavonoid family and is mainly found in plants, including fruits, green tea, vegetal vegetables, and red wine, and has antioxidant activity. Lucia et al. designed and synthesized quercetin derivatives containing triphenylphosphine and performed a preliminary biological evaluation [92]. The results showed that the compound acted as a cytotoxic agent against fast-growing cells, but not against slow-growing cells in culture, implying that TPP-quercetin has the potential to inhibit cancer cell growth. Resveratrol is a tri-phenolic compound naturally occurring in grapes and other plants that scavenges superoxide, hydroxyl radicals and metal-induced free radicals. It has been shown that mito-resveratrol, the mitochondrial targeting product of resveratrol, has relatively strong anti-proliferative activity against colon tumor cells [93]. Betulinic acid is a pentacyclic triterpenoid extracted from birch wood with antitumor, antiviral and anti-inflammatory activities. Betulinic acid has been found to induce apoptosis and inhibit tumor growth in cancer cells [94]. However, being a highly hydrophobic compound, poor blood solubility leads to limited intracellular accumulation of betulinic acid. However, when it is combined with TPP, its properties are significantly altered [95]. As it preferentially accumulates in mitochondria, it inhibits mitochondrial respiration and induces apoptosis resulting in increased bioavailability and higher cytotoxicity. A series of mitochondria-targeting analogs of betulinic acid have been synthesized and tested for their toxic effects on cancer cells [96], among them, Mito-betulinic acid has relatively strong antitumor effects on both mast cell tumor and Ehrlich Ascites carcinoma. The complex backbone structure and remarkable biological activity of natural drug molecules make them an ideal template for new drug research [97]. However, according to the current international experience and results of new drug development, there are not many examples of natural drug molecules being developed directly into new drugs, and in most cases structural modifications and optimization are required to obtain ideal new drug candidates. Structural modification of natural drug molecules through the use of TPP often improves the efficacy of the drug.

#### 4.1.3. Drugs Initially Used Outside of Cancer Treatment

Metformin is a clinical drug used to treat type 2 diabetes. Some studies have shown that patients with diabetes are more likely to develop cancer, but those treated with metformin are significantly less likely to develop cancer [98]. Due to the hydrophilic nature of metformin, it does not simply diffuse across the cell membrane, but enters the cell through uptake of transporter proteins [99]. Once metformin enters the cell, it accumulates in the mitochondrial matrix. By coupling with TPP, the mitochondrial targeting of metformin is effectively improved and its anti-tumor ability is significantly enhanced. Several experiments have shown that Mito-metformin is 100-fold more effective than free metformin in inhibiting pancreatic tumor cell proliferation in vitro and more effective in vivo. In addition, TPP-metformin-treated pancreatic tumor cells exhibited higher radiosensitivity and reduced transcription factor FOXM1. Due to the accumulation of metformin in mitochondria, the activity of electron transport chain complex I was inhibited, the rate of oxygen consumption was significantly reduced, and the production of reactive oxygen species was increased [100]. Meanwhile, activation of AMPK inhibits mTORC1 through AMPK-dependent and non-dependent mechanisms, thereby suppressing tumor cell growth [101]. In addition, the literature reports that AMPK and FOXM1D1 inhibition increases the radiosensitivity of human tumor cells [102]. Therefore, we can also consider metformin as a potential radiosensitizer. MitoQ is a targeted antioxidant used to restore mitochondrial function and improve anti-aging capacity of the body. Currently, MitoQ has been reported to be selectively toxic to tumor cells [103]. The structurally similar SKQ1 molecule also inhibited the spontaneous development of lymphomas in p53-deficient mice and the growth of human colorectal cancer xenograft HCT116 in thymus-free mice [104]. Artemisinin, a specific anti-malarial drug, has saved millions of lives [105]. Artemisinin-based combination therapies are still the first-line treatment for malaria recommended by the World Health Organization (WHO) [106]. However, recent studies have shown that ART analogs have shown limited efficacy in certain tumor cell lines, suggesting that it may also have potential as an anti-cancer drug [107]. When ART is conjugated with TPP, it has the ability to target mitochondria, and the anti-cancer effect is more obvious than that of ART [108]. Gallic acid (GA) is a well-known herbal antioxidant with various biological activities such as anti-inflammatory, antibacterial, antiviral and antitumor [109]. For over 60 years, octyl, propyl, and lauryl gallates have been permitted for use as antioxidant additives in the food and pharmaceutical industries because they have shown significantly low toxicities both in vitro and in vivo [110]. José A. Jara et al. synthesized four different (with different sizes of alkyl linkage chains) TPP-gallate compounds by chemical modification of GA [111]. The better drug molecule (TPP C_10_-GA) contains 10 carbon atoms in the linkage chain, has a significant killing effect on tumor cells and has a selectivity index for tumor cells that is 17 times higher than that of normal cells. The in vivo results also showed that TPP C_10_-GA significantly inhibited the growth of TA3/Ha tumors in mice. In 2020, the group evaluated the cytotoxicity and mode of action of another set of benzoic acid derivatives linked to TPP in colorectal cancer cell lines, including effects on metastasis and drug-resistant behavior [112]. The discovery and development of cancer drugs is a very expensive and lengthy task, often taking more than a decade and billions of dollars to prepare a drug for cancer. However, the huge amount of money invested is eventually only rewarded with a few drugs being approved for use, which has led to high prices of anti-cancer drugs to compensate for the investment in drug development. Conventional drugs have been used successfully to address other diseases and have been tested in human clinical trials with acceptable known side effects. If such drugs are combined with TPP and can exert effective antitumor effects in vivo and in vitro. This has very important implications for the research and development of antitumor drugs. As a result, the new use of conventional drugs has become an alternative strategy to overcome the huge costs and long timelines of cancer drug development.

#### 4.1.4. TPP Derivatives

Not only TPP has good antitumor effects in combination with different drug molecules, but other TPP derivatives have also shown good antitumor effects. Several alkyl derivatives of TPP have been reported to selectively inhibit the proliferation of tumor cells by inhibiting mitochondrial respiration and ATP production [113]. For example, dodecyl TPP inhibited the proliferation of suspended breast cancer stem cells in a dose-dependent manner [114]. In addition, tetraphenylphosphine cations selectively inhibited the growth of pancreatic tumor cells. However, overall, the anticancer effects of TPP derivatives were much less than those of TPP adducts with anticancer drug molecules [115].

### 4.2. TPP-Modified Targeted Mitochondrial Nanosystems

Although the strategy of directly affixing TPP with anticancer drugs has played a good mitochondrial targeting effect, the drawbacks of these affixes in terms of water solubility and biocompatibility make them still some distance away from clinical application. In contrast, strategies using nanocarriers loaded with anticancer drugs have the advantages of good biocompatibility, improved chemical stability of drugs, and modulated pharmacokinetic characteristics of drugs, and some nanocarriers can also control the release of their delivered drugs at specific target sites (as shown in Figure 6) [116].

Currently, the most widely studied mitochondrial targeting carriers include liposomes, micelles, vesicles, nanoparticles, dendrimers, gold nanoparticles, carbon nanotubes and inorganic nanocarriers such as mesoporous silica. We summarize some recent examples of nanosystems based on TPPs successfully targeting mitochondria and present them. Among the TPPs widely used for mitochondrial targeted drug delivery nanosystems, the most representative one is the coupling of docetaxel (DTX) with TPP for mitochondrial targeting by Battogtokh et al. and the loading of TPP-DTX onto folic acid-cholesterol albumin nanoparticles for anti-tumor with remarkable efficacy [117]. Liposomes are artificial membranes with bilayers that carry hydrophilic and lipophilic drugs, with the former distributed within the core compartment and the latter within the bilayer [118]. For example, peng [104] et al. prepared Lip-SPG, a multi-targeted redox-sensitive liposome co-modified with glucose and TPP, to effectively deliver DOX and the chemosensitizer lonidamine (LND) for glioma treatment (as shown in Figure 4C). In vitro experiments have shown that Lip-SPG, mediated by glucose and TPP, can target several brain regions, tumors and mitochondria, enhance the uptake of DOX and LND by mitochondria of cancer cells, and effectively inhibit tumor proliferation and induce apoptosis. As already mentioned, several natural drug molecules have antitumor activity, but their low solubility and low permeability severely inhibit their antitumor effects. Nanomicelles are easy to prepare, stable and small in size, and can significantly improve the solubility of drugs, while increasing the accumulation of drugs at tumor sites through enhanced permeation retention effect (EPR effect) and improving the effect of tumor chemotherapy [119]. Han et al. prepared a nanomicelles system TPP-PEG-PCL loaded with norcantharidin, which significantly increased the uptake of the natural drug molecule nortriptyline by tumor cells and aggregated the drug in mitochondria, enhancing antitumor efficacy by decreasing mitochondrial membrane potential, increasing intracellular ROS levels, decreasing Bax, and increasing Bcl-2 expression [120]. In addition to nanoparticles, liposomes and micelles, another commonly used nanocarrier is the dendrimer. Dendrimers can either encapsulate the drug into the core or chemically bind to the drug and attach it to the surface. Surface modification of dendrimers with different molecules can give them different properties. For example, polyethylene glycol (PEG) modification can improve solubility, linking cell recognition ligands can achieve targeting, and linking luminescent substances can be used for imaging, etc. Biswas et al. [121] grafted TPP onto the PAMAM dendrimer as a ligand for targeting mitochondria, while partial acetylation of the amino group on the surface of the dendrimer could neutralize a portion of the positive charge, which could, to some extent, avoid non-specific binding of the dendrimer to other cells and thus reduce its toxicity. Curcumin has strong anticancer activity, but the electron-dense nature of the aromatic structure significantly reduces the uptake of non-polar curcumin into the mitochondria, hindering the efficacy [122]. Kianamiri et al. [122] used a fourth-generation dendritic polymer PAMAM grafted with TPP and suffixed with curcumin for ex vivo activity evaluation. It was shown that targeted dendritic polycurcumin (TDC) could successfully deliver the drug to cancer cell mitochondria and block cell cycle G2/M. In addition, in vivo studies have shown significant tumor suppression and improved survival after TDC treatment compared to free curcumin, as well as improved drug solubility and stability. Notably, although TPP as a ligand-modified nanosystem has good mitochondrial targeting ability in tumor cells, it may form non-specific aggregation with negatively charged components in blood (e.g., plasma proteins) during the transport process, resulting in the loss of targeting ability of the transport system. To further enhance the mitochondrial targeting ability, researchers optimized the TPP targeting system by wrapping some special shells based on the tumor cell-specific structure or pH sensitivity and heat sensitivity of mitochondria (e.g., Figure 5b). It is reported that Xiao-Liang et al., [123] first experimentally synthesized charge-reversible nanorods (ZnO-TPP@D/H NRS), whose outermost layer consists of heparin shell and TPP-DOX. Due to the presence of heparin shell, the nanorods are in the anionic state during delivery, while heparanase, an effective promoter of tumor microenvironment, is specifically expressed in most tumor cells [124]. When in the vicinity of tumor cells, the heparin shell is disassembled to release TPP-DOX, which then enters the mitochondria of tumor cells driven by the membrane potential difference to exert the toxic effect of the drug on tumor cells [125].

## 5. Conclusions

Since mitochondria play an extremely important role in the regulation of biological growth and metabolism, the design of mitochondria as drug targets is of great importance in tumor therapy. In this review, we discuss common mitochondrial targeting agents with a focus on the application of TPP in mitochondrial targeted transport. As a common DLC, the lipid solubility of TPP allows it to pass easily through biological membranes, carrying a positive charge that provides it with the power needed to enter mitochondria. In addition, the mitochondrial membrane potential of tumor cells is higher than that of normal cells. This means that TPP-based targeted drugs can preferentially accumulate in tumor cells with selectivity. The advantages of TPP for mitochondrial targeted delivery are its relatively simple synthesis and purification, the ease of carrying the drug across the membrane barrier compared to other lipophilic cations and its good safety profile. Early studies synthesized some mitochondria-targeted anticancer drug based on the idea of direct adducts of TPP and anticancer drugs, which exhibit higher anticancer ability and safety than free anticancer drugs. However, they also have disadvantages such as poor water solubility and low bioavailability. Compared to traditional mitochondria-targeted delivery strategies, mitochondria-targeted nanosystems have the following advantages: Carrying traditional drugs through nanomaterials can improve drug solubility, prolong drug half-life in vivo and improve bioavailability [126]; increase the drug concentration and therapeutic index at the tumor site by aggregation in the tumor tissue through EPR effect, and reduce the adverse effects; piggybacking multiple drugs to act on tumors simultaneously and improving therapeutic effects [119].

While the current research on mitochondria-targeted anticancer drug transport system has made some progress, it still faces some options and challenges. (1) Choice of drugs: Including traditional cytotoxic drugs, natural drug molecules, newly synthesized small molecules, drugs initially used outside of cancer treatment, biomacromolecular drugs such as protein peptides and genes. (2) Choice of targets of action: Such as enzymes in the mitochondrial transport chain, mitochondrial DNA, mitochondrial membrane permeability. (3) New compounds formed by direct binding of TPP to biologically active molecules may show biological activity other than mitochondrial targeting. It has been shown that MitoQ, a mitochondria-targeted antioxidant, effectively inhibits the function of Hsp90 by reacting with TRAP1, thus exhibiting powerful anti-cancer activity [127]. This also means that when we use TPP in combination with other drug molecules, there is also a possibility that it may react with other proteins instead to make the drug less effective against cancer or even have serious adverse effects. (4) The membrane potential of some cells may change with the cellular state. Notably, the difference in potential between tumor cells and normal cells is not particularly large, and high concentrations of cation aggregation may depolarize the mitochondrial membrane potential and endanger the survival of normal cells. Therefore, future studies should explore how to optimize the TPP components to enhance their targeting and make their effects on normal cells as small as possible. (5) Based on the limited applicability of TPP as a mitochondrial targeting agent for tumor cells, it does not target all tumor cells. One study reported that in a study of in situ breast tumor cells, the membrane potential of tumor cells from a small subset of patients showed similarities to normal epithelial cells [128]. This variability may be due to the heterogeneity of tumor cells in vivo. Higher membrane potential was also not detected in leukemia, lymphoma, neuroblastoma or osteosarcoma [129]. This suggests that the TPP-based mitochondrial targeting has partially lost its specific selection for tumor cells and that TPP-targeted antitumor drugs are not applicable in this situation. (6) Although TPP has good mitochondrial targeting activity, it may form NON-specific aggregates with many negatively charged blood components during drug transport, thus losing its role. (7) Biosafety of nanocarriers; lack of uniform evaluation criteria for mitochondrial targeting nanocarriers, how to ensure the in vivo degradation and biocompatibility of nanocarrier materials and reduce their toxicity is a major challenge for the application of nanomaterials at present. Therefore, based on evidence from existing basic research, the TPP targeting system can be applied in different physiological and biochemical states by using chemical biotechnology to optimize the mitochondrial targeting agent while taking into account of the nanocarriers and clinical outcomes. While combination therapy is often more effective than monotherapy, methodological studies and safety evaluation of combination therapy are not detailed enough, such as the examination of light and sound sources, biodegradability of nanocarriers and immune rejection of drugs, as well as the lack of TPP-based mitochondrial targeting strategies for combination therapy of tumors.

In future, researchers from various fields such as pharmacy, chemistry, biology, and materials science should collaborate to explore and optimize lipophilic cation-based mitochondrial targeted drug delivery system to achieve precision therapy and improve cancer treatment.

## Figures and Tables

**Figure 1 cancers-15-00666-f001:**
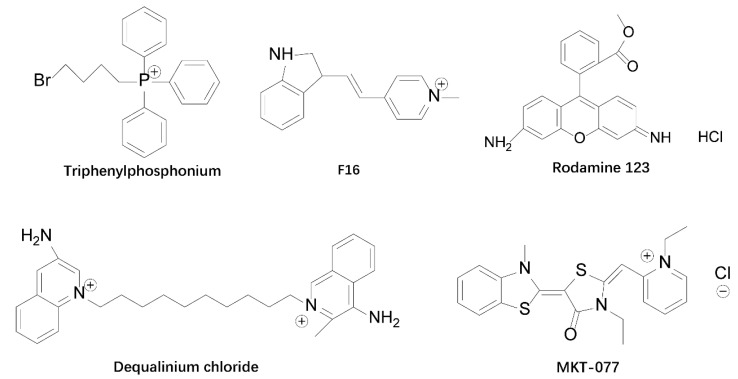
Chemical structure of some DLCs. DLCs can selectively target mitochondria driven by mitochondrial membrane potential, and they have been widely used as mitochondrial targeting carriers.

**Figure 2 cancers-15-00666-f002:**
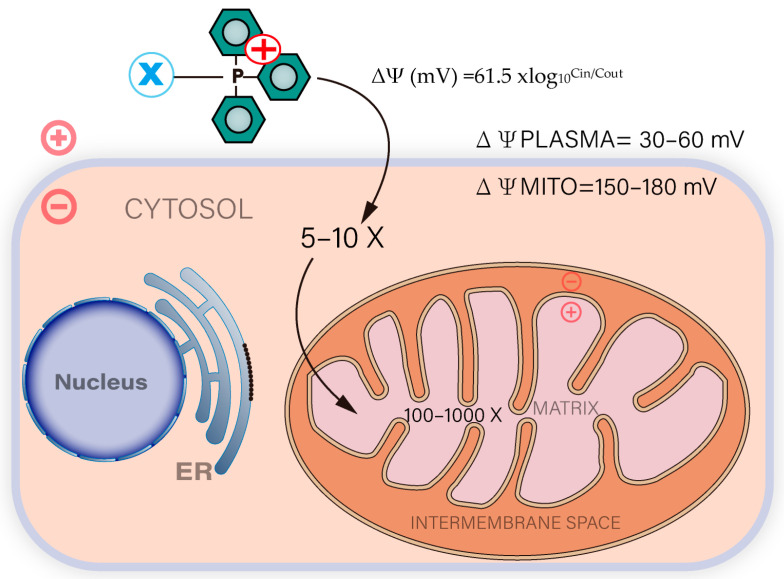
Cellular uptake of TPP conjugated compounds driven by plasma and mitochondrial membrane potentials. Compounds containing TPP can be rapidly absorbed by mitochondria driven by high mitochondrial membrane potential. At equilibrium, the ion concentrations on both sides of the charged membrane can be described by the following equation: ΔΨ (mV) = 61.5 × log_10_^Cin/Cout^.

**Figure 3 cancers-15-00666-f003:**
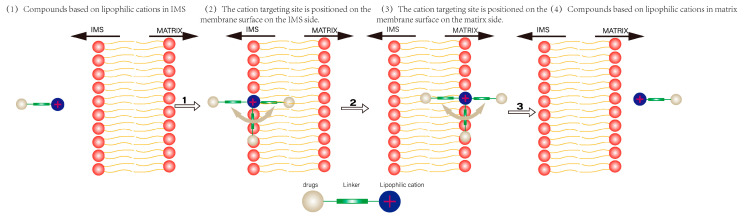
Schematic representation of the transport of a lipophilic cation carrying a drug from the mitochondrial intermembrane space (IMS) through the inner mitochondrial membrane to the mitochondrial matrix. Upon binding to the mitochondrial membrane, the cationic targeting group is localized on the membrane surface due to electrostatic interaction with negatively charged phosphate. The position of the linker and the drug will depend on their physicochemical properties. Lipophilic linkers and drugs will be localized in the center of the membrane, while hydrophilic drugs may be localized in the aqueous phase of the cell. For positively charged hydrophilic cargoes or linkers, it is possible that the molecule will “lie” on the surface of the membrane. As illustrated in Figure 3, (2) and (3).

**Figure 4 cancers-15-00666-f004:**
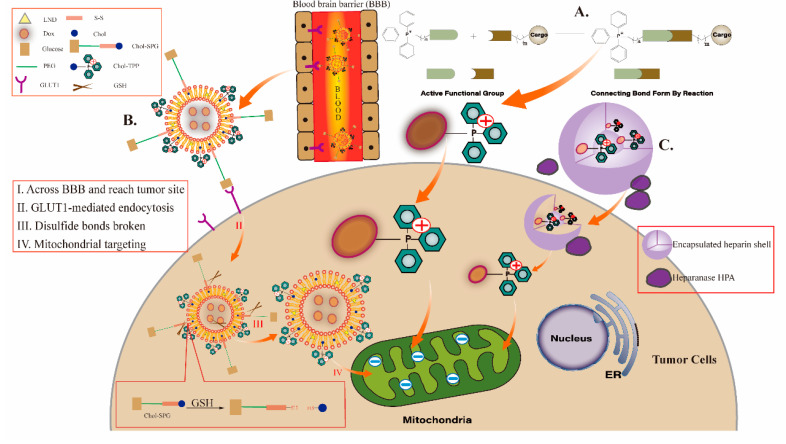
A. TPP directly conjugated with drug molecules. This is a typical molecular structure consisting of a functional part, a junction sequence and a targeting part. TPP binds to the functional part (Cargo) through a functional group reaction. B. TPP as ligand modification. For example, Lip-SPG, a multi-targeted redox liposome co-modified with glucose and TPP, can cross the BBB via GLUT1 and target tumor cell mitochondria for effective delivery of DOX and LND for glioma therapy. Chol: Cholesterol; GSH: Glutathione; LND: Lonidamine; PEG: Polyethylene glycol. C. TPP nanosystems encapsulated in special shells. For example, a heparin shell is wrapped around the surface of the drug molecule and heparanase, a tumor microenvironment promoter, is specifically expressed in most tumor cells. When located near tumor cells, the heparin shell is disassembled and the drug molecule is released to target mitochondria for action, thus avoiding the formation of non-specific aggregation of TPP with negatively charged components of the blood.

**Figure 5 cancers-15-00666-f005:**
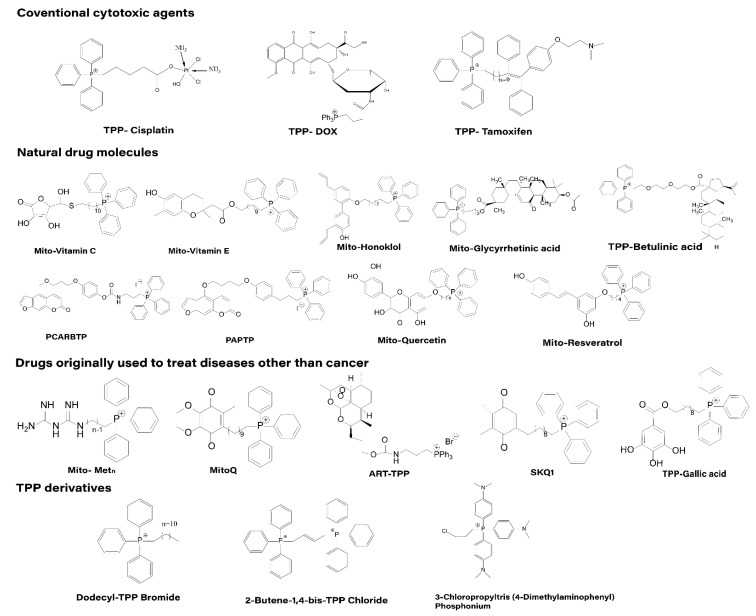
Four categories of the anticancer drugs directly conjugated to TPP: (1) Conventional cytotoxic agents, (2) natural drug molecules, (3) drugs originally used to treat diseases other than cancer, and (4) TPP derivatives.

**Figure 6 cancers-15-00666-f006:**
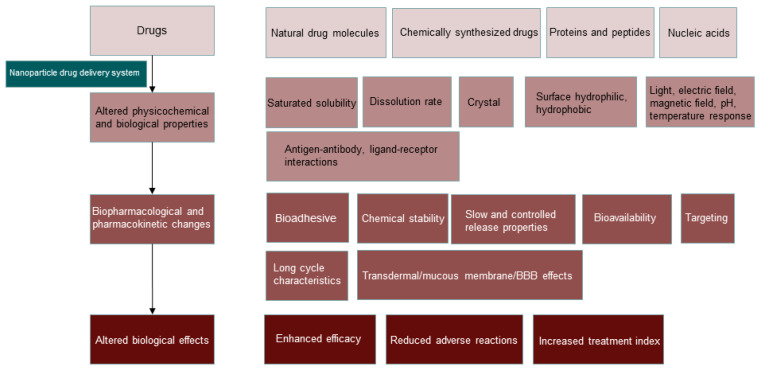
The basis of biological effects of nanomedicines. After nanosized drugs, their physical properties and biological characteristics are changed, thus affecting the distribution, metabolism and excretion of drugs, and finally achieving the purpose of enhancing drug efficacy, reducing adverse drug reactions and improving drug effects.

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
