# Peer review of "Application Prospects of Triphenylphosphine-Based Mitochondria-Targeted Cancer Therapy"

_cancers, 2023, doi:10.3390/cancers15030666_

Round 1

Reviewer 1 Report

Zhang et al review several triphenyl phosphine-based drug adducts as a mean to target selectively mitochondria in anticancer treatment. Overall the manuscript is well written and thorough. However, I have some some comments that can improve the quality of the manuscript:

1) The authors should include a section/paragraph in the manuscript where they describe physiological processes that generate a membrane potential across different organelles, including mitochondria. That section/paragraph should be in the introduction. For instance, the authors mention that ER has membrane potential. They should provide references that support this statement and also explain the mechanism that generates membrane potential in ER. 

2) line 73: the title of this section "Mitochondrial promotes" is not appropriate and misleading. The authors should change the title as the following paragraph describes the timeline for the discovery of mitochondrial targeting groups. The authors should generate a schematic representation for this timeline and include this a separate figure.

3) lines 77-79: are not clear and have to be rephrased.

4) The authors should discuss further the advantages of conjugating a drug with TPP from a mechanistic standpoint. For instance why someone should conjugate doxorubicin with mitochondria targeting molecules? What is the advantage of concentrating doxorubicin in mitochondria over the nucleus.

5) lines 198-199: the authors claim that the membrane potential of mitochondria is higher than the ER membrane potential. They should provide a reference and explain why.

6) The authors should discuss the chemical reactivity of TPP towards other proteins. For instance, the TPP of MitoQ was shown to react with TRAP1 (Yoon et al, JACS, 2021). This example should be cited as well.

7) line 248: the authors missed to put a period between the cation and the hight cLogP.

8) Figure 5: two different structures are labelled as mito resveratrol

9) line: 333: the term anti-tumor antibiotic is incorrect. 

10)The authors should explain why drug conjugates with TPP eliminate selectively cancer cells. They mention several examples that these conjugates have demonstrated a remarkable selectivity (e.g. psoralen, quercetin, gallate compounds). They should include this discussion in a different paragraph in the advantages section.

Reviewer 2 Report

Zhang and collaborators discuss the application of TPP mitochondria-target cancer therapy. The manuscript is well structured. It is detailed in some sections and in others, the authors should be more descriptive. The manuscript will benefit from accomplishing the following observations

In line 74 authors state, “Mitochondrial promoters are compounds with specific physicochemical properties 74 and reactivity”. The sentence is too vague. As the authors are detailing specific target drugs, it is essential to explain briefly which physicochemical properties are. Are they amines or ions? Do they have specific radical groups? This can be used to discuss why they have a high affinity for mitochondria. In this regard, what is the range of ki of these drugs?

In line 87, the word reinvented could be changed to another more specific as redesigned.

In line 90, specify the percentage of proteins

In line authors state, “This suggests that the function of lipophilic cat- 124 ion targeting to mitochondria is dependent on its structure”. Please discuss the specific differences among structures

In line 145, the HeH was not previously defined

The sentence in line 179 needs references

Foot figures one to three need a description.

In line 236 between [50], and as shown, there is double space.

In line 282 authors state, “TPP-based compounds are less toxic to normal cells and can be administered at high doses 282 for long periods of time without causing serious damage to organs”. Please explain in detail what long periods mean. The manuscript will benefit from including more pharmacological and toxicological terms. Please list adverse and side effects

Figures 4 and 5 need to be in higher resolution. The chemical structures seem blurred

The sentence “In addition, both TPP-DOX and HA-TPP-DOX inhibited 342 tumor cell growth in a zebrafish model with tumors, and both had no significant side 343 effects” in line 342 needs to be referenced

In line 413 among [88] and . a space is missing

In line 427, check word spaces

In line 532 authors state, “longer blood circulation time”. However, distribution and bioavailability were not further discussed. The manuscript will benefit from a table highlighting the most essential features.

Reviewer 3 Report

In this review, the authors have described and discussed mitochondria-targeted cancer therapeutic approaches, in particular the prospects of triphenylphosphine-based systems.

Overall, the review is clear, well-structured and provides the readers with both background and detailed information on the mitochondria-targeted therapy using triphenylphosphine. Most of the references are 5-15 years old, however, I do not consider it as a problem, since the review often describes the development of the methods and/or compounds. The referencing is correctly used and quite comprehensive. The figures are clear and correlate with the text.

Altogether, my recommendation is to publish the work in Cancers after minor revisions.

I have several questions/ comments which I would like to address:

 1)     the authors pay a lot of attention (several pages) to “direct affixation of TPP to drug conjugates” and not so much (one page) to “TPP-modified targeted nanosystems”. Surprisingly, then they describe how many advantages these nanosystems have over the affixes (lines 461-464; once again in the Discussion, lines 531-534). I am wondering why the paid so much less attention to the nanosystems that seem to be more suitable and prospective therapeutic agents. Maybe the authors could add several lines/ a paragraph describing/discussing the advantages of the nanosystems (lines 463-464)

2)     I also suggest that the authors might include another figure, illustrating the various nanosystems conjugated with TPP (liposomes, micelles, vesicles, dendrimers, etc.). It is not necessary but I think it may be beneficial for the readers who are not experienced in polymer/nanosystems chemistry.

3)     My other comments are minor points regarding language issues:

·        line 83: I would suggest to add “(TPP)” after the “triphenylphosphonium”.

·        lines 101-153: throughout the text describing the various lipophilic cations, I would reference immediately to the Figure 1, not at the very bottom of the page.

·        lines 111-113: should be there “…scientist studying rhodamine attached IT to antibodies…”?

·        line 125: I would use “Dequalinium chloride (DQA)” instead of “Dequalinium Chloride”.

·        line 134: I would use “DQAsomes-based” instead of “DQA-somes-based”.

·        line 176: I think the sentence miss a verb, maybe something like “…they therefore REPRESENT a reliable approach…”.

·        line 185: I would use “…from solubilization. In addition” instead of “…from solubilization; In addition”.

·        line 188: I think “inner mitochondria” is a bit confusing here. Should it be “target the inside of mitochondria” or less probably “target the inner mitochondrial membrane”?

·        line 194: I would rather use “intermembrane space” instead of “membrane gap”.

·        lines 246-248: This whole section is a bit confusing. The authors use both logP and cLogp (or even cLogP), which, to my knowledge, are slightly different units. Anyway, I think that the sentence “The cLogp of the TPP cation…” misses either a verb or any other word; it definitely misses the full stop. Could the author correct it, please?

·        line 249 and 252: The authors use the term “biofilm”. Do they really mean some macromolecular biological layers (biofilms) or biological membranes? Please, revise these.

·        line 251: I would use “such as sodium or potassium cations cannot cross” instead of “such as (Na+, K+) cannot cross”.

·        line 253: I would suggest moving the reference [52] at the end of the sentence

·        line 270: I would use “mitochondria-targeted agent” instead of “mitochondrial-targeted agent”.

·        line 292: the sentence “TPP directly from conjugations with drug molecules “ does not make much sense to me. I would use “TPP directly conjugated to/with drug molecules”.

·        line 296: There is a strange symbol after “targeting part” instead of a full stop.

·        line 297: Should be in bold both sentences highlighted in bold or just one of them? Overall, the use of bold and letters A, B, C is a bit confusing for me in the Figure 4. Personally, I like the style of the panel C, ie. “C. TPP nanosystems encapsulated in special shells.

·        line 301: “heparanase” instead of “Heparanase”.

·        line 342: I would explain the abbreviation of “HA” in the “HA-TPP-DOX”; probably it refers to the hemagglutinin (HA) tag but there is no explanation given.

·        line 427: “Gallic acid (GA) is…” contains a strange big space – is there a typo?

·        line 461: I would use a different word instead of “defects”; e.g. “problems”, “issues”, “drawbacks” or “disadvantages” etc., according to the authors´ original intentions.

·        line 518: Should it be “heparanase” instead of “heparinase”? They are different enzymes and heparanase would be consistent with its previous usage in Figure 4. Please, double check all of them and correct them, if necessary.

·        line 524: I would use “mitochondria-targeted delivery” instead of “mitochondrial-targeted delivery”.

·        line 556: I think it should be “may form NON-specific aggregates” instead of “may form specific aggregates”.

Reviewer 4 Report

The Review article from Lei Zhang et al. focuses on the potential application of TTP-based compounds that target mitochondria as anticancer treatments. The review is easy to read but a bit lengthy and wordy in some chapters (i.e. chapter 4.1.2 is too long, I would summarize a bit, also chapter 3.1 is too long, then I suggest to move the paragraph from line 248 to line 259 in the figure capture 2, see minor point 3). Overall, I believe it provides a nice state of the art of this scientific hypothesis. I have few remarks for the authors, which I feel that should be addressed in a revised version of the manuscript.

1.      The exact mechanism of action of some mitochondrial promoters and TPP-based treatments is not always explained clearly (see Line 129 for DQAsomes; line 170 for PDT-PAO-TPP; Tamoxifen-TPP mentioned in line 317). How do they affect mitochondrial ETC?

2.      In the chapter 3, lines 212-220, describing the Nernst equation is not clearly explained the contribution of proton gradient (mitochondrial pH gradient, ΔpHm), since transmembrane potential derives from the combination of proton gradient and membrane potential (ΔΨm) generated by proton pumps. This better clarified why compounds targeted to mitochondria that are weak acids or bases may undergo differential protonation in the cytosolic and mitochondrial compartments due to differences in their pH.

3.      To better explain the different classes of TTP conjugates I would suggest to add a summary Table that briefly describe them (i.e., mechanism of action, side effects, utilization in current therapies, limitations as briefly described in conclusion chapter, references, etc) to the Fig.5.

  1. There are some contradictory sentences along the manuscript that need to be better contextualized  (i.e., Line 202-205 versus line 543 and 551; line 326 versus line 332 (for example cisplatin is highly electrophilic compound so it can potentially react with nucleophilic sites of both nuclear DNA both DNA-RNA-protein of mitochondria).  

MINOR POINTS

1.      Line 13: The sentence is not clear, I suggest to write more clearly what you mean.

2.      Line 48: describe how is used nebivolol in current therapy for not expert reader, same for protopine in line 50.

3.      Figure legends should contain enough information that they should explain the whole figure content without reading the text. I would suggest to expand the figure captures as in Figure 4.

4.      Sometime it is missing the meaning of some abbreviation (i.e. LND in figure capture 4, PEG in line 495).

5.      Line 308, it should be specified which figure you are referring.

6.      Line 372 it is missing the Reference of relevance in cancer prevention, missing the Ref in line 407 and in line 434.

Round 2

Reviewer 1 Report

All of my comments were addressed.